# Learning Continuous Control Policies by Stochastic Value Gradients

**Nicolas Heess**\*, **Greg Wayne**\*, **David Silver, Timothy Lillicrap, Yuval Tassa, Tom Erez**
Google DeepMind
`{heess, gregwayne, davidsilver, countzero, tassa, etom}@google.com`

## Abstract

We present a unified framework for learning continuous control policies using backpropagation. It supports stochastic control by treating stochasticity in the Bellman equation as a deterministic function of exogenous noise. The product is a spectrum of general policy gradient algorithms that range from model-free methods with value functions to model-based methods without value functions. We use learned models but only require observations from the environment instead of observations from model-predicted trajectories, minimizing the impact of compounded model errors. We apply these algorithms first to a toy stochastic control problem and then to several physics-based control problems in simulation. One of these variants, SVG(1), shows the effectiveness of learning models, value functions, and policies simultaneously in continuous domains.

## 1 Introduction

Policy gradient algorithms maximize the expectation of cumulative reward by following the gradient of this expectation with respect to the policy parameters. Most existing algorithms estimate this gradient in a model-free manner by sampling returns from the real environment and rely on a likelihood ratio estimator [32, 26]. Such estimates tend to have high variance and require large numbers of samples or, conversely, low-dimensional policy parameterizations.

A second approach to estimate a policy gradient relies on backpropagation instead of likelihood ratio methods. If a differentiable environment model is available, one can link together the policy, model, and reward function to compute an analytic policy gradient by backpropagation of reward along a trajectory [18, 11, 6, 9]. Instead of using entire trajectories, one can estimate future rewards using a learned value function (a critic) and compute policy gradients from subsequences of trajectories. It is also possible to backpropagate analytic action derivatives from a Q-function to compute the policy gradient without a model [31, 21, 23]. Following Fairbank [8], we refer to methods that compute the policy gradient through backpropagation as *value gradient* methods.

In this paper, we address two limitations of prior value gradient algorithms. The first is that, in contrast to likelihood ratio methods, value gradient algorithms are only suitable for training deterministic policies. Stochastic policies have several advantages: for example, they can be beneficial for partially observed problems [24]; they permit on-policy exploration; and because stochastic policies can assign probability mass to off-policy trajectories, we can train a stochastic policy on samples from an experience database in a principled manner. When an environment model is used, value gradient algorithms have also been critically limited to operation in deterministic environments. By exploiting a mathematical tool known as "re-parameterization" that has found recent use for generative models [20, 12], we extend the scope of value gradient algorithms to include the optimization of stochastic policies in stochastic environments. We thus describe our framework as *Stochastic Value Gradient* (SVG) methods. Secondly, we show that an environment dynamics model, value function, and policy can be learned jointly with neural networks based only on environment interaction. Learned dynamics models are often inaccurate, which we mitigate by computing value gradients along real system trajectories instead of planned ones, a feature shared by model-free

methods [32, 26]. This substantially reduces the impact of model error because we only use models to compute policy gradients, not for prediction, combining advantages of model-based and model-free methods with fewer of their drawbacks.

We present several algorithms that range from model-based to model-free methods, flexibly combining models of environment dynamics with value functions to optimize policies in stochastic or deterministic environments. Experimentally, we demonstrate that SVG methods can be applied using generic neural networks with tens of thousands of parameters while making minimal assumptions about plants or environments. By examining a simple stochastic control problem, we show that SVG algorithms can optimize policies where model-based planning and likelihood ratio methods cannot. We provide evidence that value function approximation can compensate for degraded models, demonstrating the increased robustness of SVG methods over model-based planning. Finally, we use SVG algorithms to solve a variety of challenging, under-actuated, physical control problems, including swimming of snakes, reaching, tracking, and grabbing with a robot arm, fall-recovery for a monoped, and locomotion for a planar cheetah and biped.

## 2 Background

We consider discrete-time Markov Decision Processes (MDPs) with continuous states and actions and denote the state and action at time step $t$ by $\mathbf{s}^t \in \mathbb{R}^{N_S}$ and $\mathbf{a}^t \in \mathbb{R}^{N_A}$, respectively. The MDP has an initial state distribution $\mathbf{s}^0 \sim p^0(\cdot)$, a transition distribution $\mathbf{s}^{t+1} \sim p(\cdot|\mathbf{s}^t, \mathbf{a}^t)$, and a (potentially time-varying) reward function $r^t = r(\mathbf{s}^t, \mathbf{a}^t, t)$.[1] We consider time-invariant stochastic policies $\mathbf{a} \sim p(\cdot|\mathbf{s}; \theta)$, parameterized by $\theta$. The goal of policy optimization is to find policy parameters $\theta$ that maximize the expected sum of future rewards. We optimize either finite-horizon or infinite-horizon sums, i.e., $J(\theta) = \mathbb{E}\left[\sum_{t=0}^T \gamma^t r^t | \theta\right]$ or $J(\theta) = \mathbb{E}\left[\sum_{t=0}^\infty \gamma^t r^t | \theta\right]$ where $\gamma \in [0, 1]$ is a discount factor.[2] When possible, we represent a variable at the next time step using the "tick" notation, e.g., $\mathbf{s}' \triangleq \mathbf{s}^{t+1}$.

In what follows, we make extensive use of the state-action-value Q-function and state-value V-function.

$$Q^t(\mathbf{s}, \mathbf{a}) = \mathbb{E}\left[\sum_{\tau=t} \gamma^{\tau-t} r^\tau | \mathbf{s}^t = \mathbf{s}, \mathbf{a}^t = \mathbf{a}, \theta\right]; V^t(\mathbf{s}) = \mathbb{E}\left[\sum_{\tau=t} \gamma^{\tau-t} r^\tau | \mathbf{s}^t = \mathbf{s}, \theta\right]. \quad (1)$$

For finite-horizon problems, the value functions are time-dependent, e.g., $V' \triangleq V^{t+1}(\mathbf{s}')$, and for infinite-horizon problems the value functions are stationary, $V' \triangleq V(\mathbf{s}')$. The relevant meaning should be clear from the context. The state-value function can be expressed recursively using the stochastic Bellman equation

$$V^t(\mathbf{s}) = \int \left[r^t + \gamma \int V^{t+1}(\mathbf{s}') p(\mathbf{s}'|\mathbf{s}, \mathbf{a}) d\mathbf{s}'\right] p(\mathbf{a}|\mathbf{s}; \theta) d\mathbf{a}. \quad (2)$$

We abbreviate partial differentiation using subscripts, $g_x \triangleq \partial g(x, y)/\partial x$.

## 3 Deterministic value gradients

The deterministic Bellman equation takes the form $V(\mathbf{s}) = r(\mathbf{s}, \mathbf{a}) + \gamma V'(\mathbf{f}(\mathbf{s}, \mathbf{a}))$ for a deterministic model $\mathbf{s}' = \mathbf{f}(\mathbf{s}, \mathbf{a})$ and deterministic policy $\mathbf{a} = \pi(\mathbf{s}; \theta)$. Differentiating the equation with respect to the state and policy yields an expression for the value gradient

$$V_\mathbf{s} = r_\mathbf{s} + r_\mathbf{a}\pi_\mathbf{s} + \gamma V'_{\mathbf{s}'}(\mathbf{f}_\mathbf{s} + \mathbf{f}_\mathbf{a}\pi_\mathbf{s}), \quad (3)$$

$$V_\theta = r_\mathbf{a}\pi_\theta + \gamma V'_{\mathbf{s}'}\mathbf{f}_\mathbf{a}\pi_\theta + \gamma V'_\theta. \quad (4)$$

In eq. 4, the term $\gamma V'_\theta$ arises because the total derivative includes policy gradient contributions from subsequent time steps (full derivation in Appendix A). For a purely model-based formalism, these equations are used as a pair of coupled recursions that, starting from the termination of a trajectory, proceed backward in time to compute the gradient of the value function with respect to the state and policy parameters. $V_\theta^0$ returns the total policy gradient. When a state-value function is used

after one step in the recursion, $r_{\mathbf{a}}\pi_\theta + \gamma V'_{\mathbf{s}'}\mathbf{f}_{\mathbf{a}}\pi_\theta$ directly expresses the contribution of the current time step to the policy gradient. Summing these gradients over the trajectory gives the total policy gradient. When a Q-function is used, the per-time step contribution to the policy gradient takes the form $Q_{\mathbf{a}}\pi_\theta$.

## 4 Stochastic value gradients

One limitation of the gradient computation in eqs. 3 and 4 is that the model and policy must be deterministic. Additionally, the accuracy of the policy gradient $V_\theta$ is highly sensitive to modeling errors. We introduce two critical changes: First, in section 4.1, we transform the stochastic Bellman equation (eq. 2) to permit backpropagating value information in a stochastic setting. This also enables us to compute gradients along real trajectories, not ones sampled from a model, making the approach robust to model error, leading to our first algorithm "SVG($\infty$)," described in section 4.2. Second, in section 4.3, we show how value function critics can be integrated into this framework, leading to the algorithms "SVG(1)" and "SVG(0)", which expand the Bellman recursion for 1 and 0 steps, respectively. Value functions further increase robustness to model error and extend our framework to infinite-horizon control.

### 4.1 Differentiating the stochastic Bellman equation

**Re-parameterization of distributions** Our goal is to backpropagate through the stochastic Bellman equation. To do so, we make use of a concept called "re-parameterization", which permits us to compute derivatives of deterministic and stochastic models in the same way. A very simple example of re-parameterization is to write a conditional Gaussian density $p(y|x) = \mathcal{N}(y|\mu(x), \sigma^2(x))$ as the function $y = \mu(x) + \sigma(x)\xi$, where $\xi \sim \mathcal{N}(0,1)$. From this point of view, one produces samples procedurally by first sampling $\xi$, then deterministically constructing $y$. Here, we consider conditional densities whose samples are generated by a deterministic function of an input noise variable and other conditioning variables: $\mathbf{y} = \mathbf{f}(\mathbf{x}, \xi)$, where $\xi \sim \rho(\cdot)$, a fixed noise distribution. Rich density models can be expressed in this form [20, 12]. Expectations of a function $\mathbf{g}(\mathbf{y})$ become $\mathbb{E}_{p(\mathbf{y}|\mathbf{x})}\mathbf{g}(\mathbf{y}) = \int \mathbf{g}(\mathbf{f}(\mathbf{x}, \xi))\rho(\xi)d\xi$.

The advantage of working with re-parameterized distributions is that we can now obtain a simple Monte-Carlo estimator of the derivative of an expectation with respect to $\mathbf{x}$:

$$\nabla_{\mathbf{x}}\mathbb{E}_{p(\mathbf{y}|\mathbf{x})}\mathbf{g}(\mathbf{y}) = \mathbb{E}_{\rho(\xi)}\mathbf{g}_{\mathbf{y}}\mathbf{f}_{\mathbf{x}} \approx \frac{1}{M}\sum_{i=1}^{M}\mathbf{g}_{\mathbf{y}}\mathbf{f}_{\mathbf{x}}\big|_{\xi=\xi_i}. \tag{5}$$

In contrast to likelihood ratio-based Monte Carlo estimators, $\nabla_{\mathbf{x}}\log p(\mathbf{y}|\mathbf{x})\mathbf{g}(\mathbf{y})$, this formula makes direct use of the Jacobian of $\mathbf{g}$.

**Re-parameterization of the Bellman equation** We now re-parameterize the Bellman equation. When re-parameterized, the stochastic policy takes the form $\mathbf{a} = \pi(\mathbf{s}, \eta; \theta)$, and the stochastic environment the form $\mathbf{s}' = \mathbf{f}(\mathbf{s}, \mathbf{a}, \xi)$ for noise variables $\eta \sim \rho(\eta)$ and $\xi \sim \rho(\xi)$, respectively. Inserting these functions into eq. (2) yields

$$V(\mathbf{s}) = \mathbb{E}_{\rho(\eta)}\Big[r(\mathbf{s}, \pi(\mathbf{s}, \eta; \theta)) + \gamma\mathbb{E}_{\rho(\xi)}\big[V'(f(\mathbf{s}, \pi(\mathbf{s}, \eta; \theta), \xi))\big]\Big]. \tag{6}$$

Differentiating eq. 6 with respect to the current state $\mathbf{s}$ and policy parameters $\theta$ gives

$$V_{\mathbf{s}} = \mathbb{E}_{\rho(\eta)}\Big[r_{\mathbf{s}} + r_{\mathbf{a}}\pi_{\mathbf{s}} + \gamma\mathbb{E}_{\rho(\xi)}V'_{\mathbf{s}'}(\mathbf{f}_{\mathbf{s}} + \mathbf{f}_{\mathbf{a}}\pi_{\mathbf{s}})\Big], \tag{7}$$

$$V_{\theta} = \mathbb{E}_{\rho(\eta)}\Big[r_{\mathbf{a}}\pi_{\theta} + \gamma\mathbb{E}_{\rho(\xi)}\big[V'_{\mathbf{s}'}\mathbf{f}_{\mathbf{a}}\pi_{\theta} + V'_{\theta}\big]\Big]. \tag{8}$$

We are interested in controlling systems with *a priori* unknown dynamics. Consequently, in the following, we replace instances of $\mathbf{f}$ or its derivatives with a learned model $\hat{\mathbf{f}}$.

**Gradient evaluation by planning** A planning method to compute a gradient estimate is to compute a trajectory by running the policy in loop with a model while sampling the associated noise variables, yielding a trajectory $\tau = (\mathbf{s}^1, \eta^1, \mathbf{a}^1, \xi^1, \mathbf{s}^2, \eta^2, \mathbf{a}^2, \xi^2, \dots)$. On this sampled trajectory, a Monte-Carlo estimate of the policy gradient can be computed by the backward recursions:

$$v_{\mathbf{s}} = [r_{\mathbf{s}} + r_{\mathbf{a}}\pi_{\mathbf{s}} + \gamma v'_{\mathbf{s}'}(\hat{\mathbf{f}}_{\mathbf{s}} + \hat{\mathbf{f}}_{\mathbf{a}}\pi_{\mathbf{s}})]\big|_{\eta,\xi}, \tag{9}$$

$$v_{\theta} = [r_{\mathbf{a}}\pi_{\theta} + \gamma(v'_{\mathbf{s}'}\hat{\mathbf{f}}_{\mathbf{a}}\pi_{\theta} + v'_{\theta})]\big|_{\eta,\xi}, \tag{10}$$

where have written lower-case $v$ to emphasize that the quantities are one-sample estimates[3], and "$\big|_{x}$" means "evaluated at $x$".

**Gradient evaluation on real trajectories**  An important advantage of stochastic over deterministic models is that they can assign probability mass to observations produced by the real environment. In a deterministic formulation, there is no principled way to account for mismatch between model predictions and observed trajectories. In this case, the policy and environment noise $(\eta, \xi)$ that produced the observed trajectory are considered unknown. By an application of Bayes' rule, which we explain in Appendix B, we can rewrite the expectations in equations 7 and 8 given the observations $(\mathbf{s}, \mathbf{a}, \mathbf{s}')$ as

$$V_{\mathbf{s}} = \mathbb{E}_{p(\mathbf{a}|\mathbf{s})}\mathbb{E}_{p(\mathbf{s}'|\mathbf{s},\mathbf{a})}\mathbb{E}_{p(\eta,\xi|\mathbf{s},\mathbf{a},\mathbf{s}')}\left[r_{\mathbf{s}} + r_{\mathbf{a}}\pi_{+}\gamma V'_{\mathbf{s}'}(\hat{\mathbf{f}}_{\mathbf{s}} + \hat{\mathbf{f}}_{\mathbf{a}}\pi_{\mathbf{s}})\right], \tag{11}$$

$$V_{\theta} = \mathbb{E}_{p(\mathbf{a}|\mathbf{s})}\mathbb{E}_{p(\mathbf{s}'|\mathbf{s},\mathbf{a})}\mathbb{E}_{p(\eta,\xi|\mathbf{s},\mathbf{a},\mathbf{s}')}\left[r_{\mathbf{a}}\pi_{\theta} + \gamma(V'_{\mathbf{s}'}\hat{\mathbf{f}}_{\mathbf{a}}\pi_{\theta} + V'_{\theta})\right], \tag{12}$$

where we can now replace the two outer expectations with samples derived from interaction with the real environment. In the special case of additive noise, $\mathbf{s}' = \hat{\mathbf{f}}(\mathbf{s}, \mathbf{a}) + \xi$, it is possible to use a deterministic model to compute the derivatives $(\hat{\mathbf{f}}_{\mathbf{s}}, \hat{\mathbf{f}}_{\mathbf{a}})$. The noise's influence is restricted to the gradient of the value of the next state, $V'_{\mathbf{s}'}$, and does not affect the model Jacobian. If we consider it desirable to capture more complicated environment noise, we can use a re-parameterized generative model and infer the missing noise variables, possibly by sampling from $p(\eta, \xi|\mathbf{s}, \mathbf{a}, \mathbf{s}')$.

## 4.2  SVG($\infty$)

SVG($\infty$) computes value gradients by backward recursions on finite-horizon trajectories. After every episode, we train the model, $\hat{\mathbf{f}}$, followed by the policy, $\pi$. We provide pseudocode for this in Algorithm 1 but discuss further implementation details in section 5 and in the experiments.

---

**Algorithm 1** SVG($\infty$)

1: Given empty experience database $\mathcal{D}$
2: **for** trajectory $= 0$ **to** $\infty$ **do**
3:   **for** $t = 0$ **to** $T$ **do**
4:     Apply control $\mathbf{a} = \pi(\mathbf{s}, \eta; \theta), \eta \sim \rho(\eta)$
5:     Insert $(\mathbf{s}, \mathbf{a}, r, \mathbf{s}')$ into $\mathcal{D}$
6:   **end for**
7:   Train generative model $\hat{\mathbf{f}}$ using $\mathcal{D}$
8:   $v'_{\mathbf{s}} = 0$ (finite-horizon)
9:   $v'_{\theta} = 0$ (finite-horizon)
10:   **for** $t = T$ down to $0$ **do**
11:     Infer $\xi|(\mathbf{s}, \mathbf{a}, \mathbf{s}')$ and $\eta|(\mathbf{s}, \mathbf{a})$
12:     $v_{\theta} = [r_{\mathbf{a}}\pi_{\theta} + \gamma(v'_{\mathbf{s}'}\hat{\mathbf{f}}_{\mathbf{a}}\pi_{\theta} + v'_{\theta})]\big|_{\eta,\xi}$
13:     $v_{\mathbf{s}} = [r_{\mathbf{s}} + r_{\mathbf{a}}\pi_{\mathbf{s}} + \gamma v'_{\mathbf{s}'}(\hat{\mathbf{f}}_{\mathbf{s}} + \hat{\mathbf{f}}_{\mathbf{a}}\pi_{\mathbf{s}})]\big|_{\eta,\xi}$
14:   **end for**
15:   Apply gradient-based update using $v^0_{\theta}$
16: **end for**

**Algorithm 2** SVG(1) with Replay

1: Given empty experience database $\mathcal{D}$
2: **for** $t = 0$ **to** $\infty$ **do**
3:   Apply control $\pi(\mathbf{s}, \eta; \theta), \eta \sim \rho(\eta)$
4:   Observe $r, \mathbf{s}'$
5:   Insert $(\mathbf{s}, \mathbf{a}, r, \mathbf{s}')$ into $\mathcal{D}$
6:   // Model and critic updates
7:   Train generative model $\hat{\mathbf{f}}$ using $\mathcal{D}$
8:   Train value function $\hat{V}$ using $\mathcal{D}$ (Alg. 4)
9:   // Policy update
10:   Sample $(\mathbf{s}^k, \mathbf{a}^k, r^k, \mathbf{s}^{k+1})$ from $\mathcal{D}$ ($k \leq t$)
11:   $w = \frac{p(\mathbf{a}^k|\mathbf{s}^k; \theta^t)}{p(\mathbf{a}^k|\mathbf{s}^k; \theta^k)}$
12:   Infer $\xi^k|(\mathbf{s}^k, \mathbf{a}^k, \mathbf{s}^{k+1})$ and $\eta^k|(\mathbf{s}^k, \mathbf{a}^k)$
13:   $v_{\theta} = w(r_{\mathbf{a}} + \gamma \hat{V}'_{\mathbf{s}'}\hat{\mathbf{f}}_{\mathbf{a}})\pi_{\theta}\big|_{\eta^k, \xi^k}$
14:   Apply gradient-based update using $v_{\theta}$
15: **end for**

---

## 4.3  SVG(1) and SVG(0)

In our framework, we may learn a parametric estimate of the expected value $\hat{V}(\mathbf{s}; \nu)$ (critic) with parameters $\nu$. The derivative of the critic value with respect to the state, $\hat{V}_{\mathbf{s}}$, can be used in place of the sample gradient estimate given in eq. (9). The critic can reduce the variance of the gradient estimates because $\hat{V}$ approximates the *expectation* of future rewards while eq. (9) provides only a

single-trajectory estimate. Additionally, the value function can be used at the end of an episode to approximate the infinite-horizon policy gradient. Finally, eq. (9) involves the repeated multiplication of Jacobians of the approximate model $\hat{\mathbf{f}}_{\mathbf{s}}$, $\hat{\mathbf{f}}_{\mathbf{a}}$. Just as model error can compound in forward planning, model gradient error can compound during backpropagation. Furthermore, SVG($\infty$) is on-policy. That is, after each episode, a single gradient-based update is made to the policy, and the policy optimization does not revisit those trajectory data again. To increase data-efficiency, we construct an off-policy, experience replay [15, 29] algorithm that uses models and value functions, SVG(1) with Experience Replay (SVG(1)-ER). This algorithm also has the advantage that it can perform an infinite-horizon computation.

To construct an off-policy estimator, we perform importance-weighting of the current policy distribution with respect to a proposal distribution, $q(\mathbf{s}, \mathbf{a})$:

$$\hat{V}_\theta = \mathbb{E}_{q(\mathbf{s}, \mathbf{a})} \mathbb{E}_{p(\mathbf{s}'|\mathbf{s}, \mathbf{a})} \mathbb{E}_{p(\eta, \xi|\mathbf{s}, \mathbf{a}, \mathbf{s}')} \frac{p(\mathbf{a}|\mathbf{s}; \theta)}{q(\mathbf{a}|\mathbf{s})} \left[ r_{\mathbf{a}} \pi_\theta + \gamma \hat{V}'_{\mathbf{s}} \hat{\mathbf{f}}_{\mathbf{a}} \pi_\theta \right]. \tag{13}$$

Specifically, we maintain a database with tuples of past state transitions $(\mathbf{s}^k, \mathbf{a}^k, r^k, \mathbf{s}^{k+1})$. Each proposal drawn from $q$ is a sample of a tuple from the database. At time $t$, the importance-weight $w \triangleq p/q = \frac{p(\mathbf{a}^k|\mathbf{s}^k; \theta^t)}{p(\mathbf{a}^k|\mathbf{s}^k, \theta^k)}$, where $\theta^k$ comprise the policy parameters in use at the historical time step $k$. We do not importance-weight the marginal distribution over states $q(\mathbf{s})$ generated by a policy; this is widely considered to be intractable.

Similarly, we use experience replay for value function learning. Details can be found in Appendix C. Pseudocode for the SVG(1) algorithm with Experience Replay is in Algorithm 2.

We also provide a model-free stochastic value gradient algorithm, SVG(0) (Algorithm 3 in the Appendix). This algorithm is very similar to SVG(1) and is the stochastic analogue of the recently introduced Deterministic Policy Gradient algorithm (DPG) [23, 14, 4]. Unlike DPG, instead of assuming a deterministic policy, SVG(0) estimates the derivative around the policy noise $\mathbb{E}_{p(\eta)} \left[ Q_{\mathbf{a}} \pi_\theta | \eta \right]$.[4] This, for example, permits learning policy noise variance. The relative merit of SVG(1) versus SVG(0) depends on whether the model or value function is easier to learn and is task-dependent. We expect that model-based algorithms such as SVG(1) will show the strongest advantages in multitask settings where the system dynamics are fixed, but the reward function is variable. SVG(1) performed well across all experiments, including ones introducing capacity constraints on the value function and model. SVG(1)-ER demonstrated a significant advantage over all other tested algorithms.

## 5   Model and value learning

We can use almost any kind of differentiable, generative model. In our work, we have parameterized the models as neural networks. Our framework supports nonlinear state- and action-dependent noise, notable properties of biological actuators. For example, this can be described by the parametric form $\hat{\mathbf{f}}(\mathbf{s}, \mathbf{a}, \xi) = \hat{\mu}(\mathbf{s}, \mathbf{a}) + \hat{\sigma}(\mathbf{s}, \mathbf{a}) \xi$. Model learning amounts to a purely supervised problem based on observed state transitions. Our model and policy training occur *jointly*. There is no "motor-babbling" period used to identify the model. As new transitions are observed, the model is trained first, followed by the value function (for SVG(1)), followed by the policy. To ensure that the model does not forget information about state transitions, we maintain an experience database and cull batches of examples from the database for every model update. Additionally, we model the state-change by $\mathbf{s}' = \hat{\mathbf{f}}(\mathbf{s}, \mathbf{a}, \xi) + \mathbf{s}$ and have found that constructing models as separate sub-networks per predicted state dimension improved model quality significantly.

Our framework also permits a variety of means to learn the value function models. We can use temporal difference learning [25] or regression to empirical episode returns. Since SVG(1) is model-based, we can also use Bellman residual minimization [3]. In practice, we used a version of "fitted" policy evaluation. Pseudocode is available in Appendix C, Algorithm 4.

## 6   Experiments

We tested the SVG algorithms in two sets of experiments. In the first set of experiments (section 6.1), we test whether evaluating gradients on real environment trajectories and value function ap-

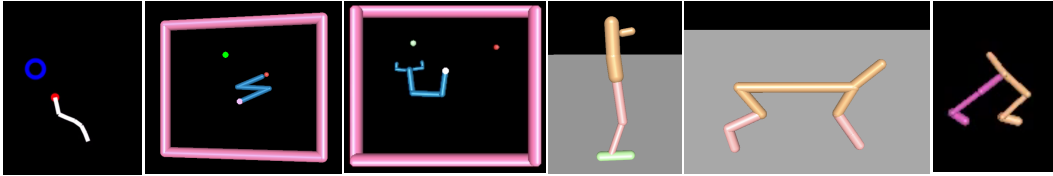

Figure 1: *From left to right*: 7-Link Swimmer; Reacher; Gripper; Monoped; Half-Cheetah; Walker

proximation can reduce the impact of model error. In our second set (section 6.2), we show that SVG(1) can be applied to several complicated, multidimensional physics environments involving contact dynamics (Figure 1) in the MuJoCo simulator [28]. Below we only briefly summarize the main properties of each environment: further details of the simulations can be found in Appendix D and supplement. In all cases, we use generic, 2 hidden-layer neural networks with *tanh* activation functions to represent models, value functions, and policies. A video montage is available at `https://youtu.be/PYdL7bcn_cM`.

## 6.1 Analyzing SVG

**Gradient evaluation on real trajectories vs. planning**    To demonstrate the difficulty of planning with a stochastic model, we first present a very simple control problem for which SVG($\infty$) easily learns a control policy but for which an otherwise identical planner fails entirely. Our example is based on a problem due to [16]. The policy directly controls the velocity of a point-mass "hand" on a 2D plane. By means of a spring-coupling, the hand exerts a force on a ball mass; the ball additionally experiences a gravitational force and random forces (Gaussian noise). The goal is to bring hand and ball into one of two randomly chosen target configurations with a relevant reward being provided only at the final time step.

With simulation time step $0.01s$, this demands controlling and backpropagating the distal reward along a trajectory of $1,000$ steps. Because this experiment has a non-stationary, time-dependent value function, this problem also favors model-based value gradients over methods using value functions. SVG($\infty$) easily learns this task, but the planner, which uses trajectories from the model, shows little improvement. The planner simulates trajectories using the learned stochastic model and backpropagates along those simulated trajectories (eqs. 9 and 10) [18]. The extremely long time-horizon lets prediction error accumulate and thus renders roll-outs highly inaccurate, leading to much worse final performance (c.f. Fig. 2, *left*).[5]

**Robustness to degraded models and value functions**    We investigated the sensitivity of SVG($\infty$) and SVG(1) to the quality of the learned model on Swimmer. Swimmer is a chain body with multiple links immersed in a fluid environment with drag forces that allow the body to propel itself [5, 27]. We build chains of 3, 5, or 7 links, corresponding to 10, 14, or 18-dimensional state spaces with 2, 4, or 6-dimensional action spaces. The body is initialized in random configurations with respect to a central goal location. Thus, to solve the task, the body must turn to re-orient and then produce an undulation to move to the goal.

To assess the impact of model quality, we learned to control a link-3 swimmer with SVG($\infty$) and SVG(1) while varying the capacity of the network used to model the environment (5, 10, or 20 hidden units for each state dimension subnetwork (Appendix D); i.e., in this task we intentionally shrink the neural network model to investigate the sensitivity of our methods to model inaccuracy. While with a high capacity model (20 hidden units per state dimension), both SVG($\infty$) and SVG(1) successfully learn to solve the task, the performance of SVG($\infty$) drops significantly as model capacity is reduced (c.f. Fig. 3, *middle*). SVG(1) still works well for models with only 5 hidden units, and it also scales up to 5 and 7-link versions of the swimmer (Figs. 3, *right* and 4, *left*). To compare SVG(1) to conventional model-free approaches, we also tested a state-of-the-art actor-critic algorithm that learns a $V$-function and updates the policy using the TD-error $\delta = r + \gamma V' - V$ as an estimate of the advantage, yielding the policy gradient $v_\theta = \delta \nabla_\theta \log \pi$ [30]. (SVG(1) and the AC algorithm used the same code for learning $V$.) SVG(1) outperformed the model-free approach in the 3-, 5-, and 7-link swimmer tasks (c.f. Fig. 3, *left*, *right*; Fig. 4, *top left*). In figure panels 2, *middle*, 3, *right*, and 4, *left column*, we show that experience replay for the policy can improve the data efficiency and performance of SVG(1).

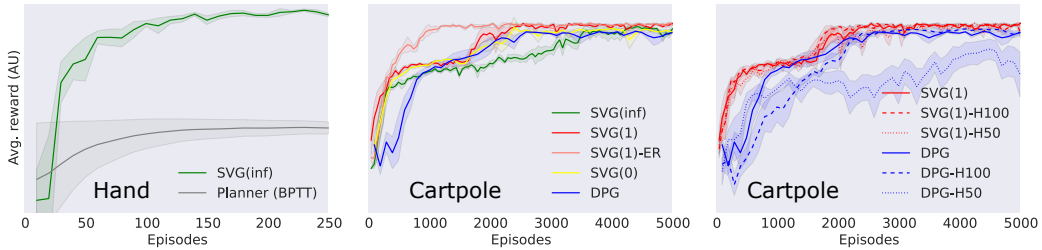

Figure 2: *Left*: Backpropagation through a model along observed stochastic trajectories is able to optimize a stochastic policy in a stochastic environment, but an otherwise equivalent planning algorithm that simulates the transitions with a learned stochastic model makes little progress due to compounding model error. *Middle*: SVG and DPG algorithms on cart-pole. SVG(1)-ER learns the fastest. *Right*: When the value function capacity is reduced from 200 hidden units in the first layer to 100 and then again to 50, SVG(1) exhibits less performance degradation than the Q-function-based DPG, presumably because the dynamics model contains auxiliary information about the Q function.

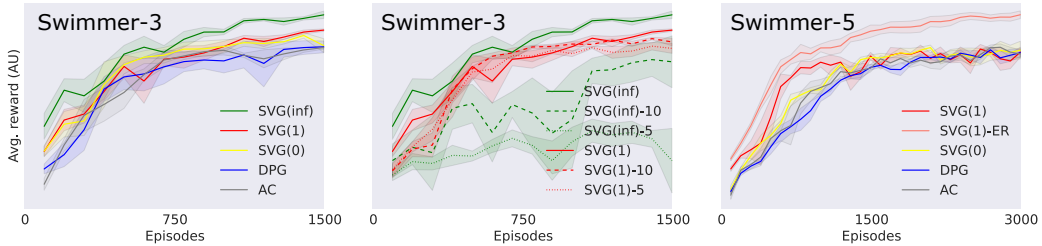

Figure 3: *Left*: For a 3-link swimmer, with relatively simple dynamics, the compared methods yield similar results and possibly a slight advantage to the purely model-based SVG($\infty$). *Middle*: However, as the environment model's capacity is reduced from 20 to 10 then to 5 hidden units per state-dimension subnetwork, SVG($\infty$) dramatically deteriorates, whereas SVG(1) shows undisturbed performance. *Right*: For a 5-link swimmer, SVG(1)-ER learns faster and asymptotes at higher performance than the other tested algorithms.

Similarly, we tested the impact of varying the capacity of the value function approximator (Fig. 2, *right*) on a cart-pole. The V-function-based SVG(1) degrades less severely than the Q-function-based DPG presumably because it computes the policy gradient with the aid of the dynamics model.

## 6.2 SVG in complex environments

In a second set of experiments we demonstrated that SVG(1)-ER can be applied to several challenging physical control problems with stochastic, non-linear, and discontinuous dynamics due to contacts. *Reacher* is an arm stationed within a walled box with 6 state dimensions and 3 action dimensions and the $(x, y)$ coordinates of a target site, giving 8 state dimensions in total. In 4-Target Reacher, the site was randomly placed at one of the four corners of the box, and the arm in a random configuration at the beginning of each trial. In Moving-Target Reacher, the site moved at a randomized speed and heading in the box with reflections at the walls. Solving this latter problem implies that the policy has generalized over the entire work space. *Gripper* augments the reacher arm with a manipulator that can grab a ball in a randomized position and return it to a specified site. *Monoped* has 14 state dimensions, 4 action dimensions, and ground contact dynamics. The monoped begins falling from a height and must remain standing. Additionally, we apply Gaussian random noise to the torques controlling the joints with a standard deviation of $5\%$ of the total possible actuator strength at all points in time, reducing the stability of upright postures. *Half-Cheetah* is a planar cat robot designed to run based on [29] with 18 state dimensions and 6 action dimensions. Half-Cheetah has a version with springs to aid balanced standing and a version without them. *Walker* is a planar biped, based on the environment from [22].

**Results** Figure 4 shows learning curves for several repeats for each of the tasks. We found that in all cases SVG(1) solved the problem well; we provide videos of the learned policies in the supplemental material. The 4-target reacher reliably finished at the target site, and in the tracking task followed the moving target successfully. SVG(1)-ER has a clear advantage on this task as also borne out in the cart-pole and swimmer experiments. The cheetah gaits varied slightly from experiment to experiment but in all cases made good forward progress. For the monoped, the policies were able to balance well beyond the 200 time steps of training episodes and were able to resist significantly

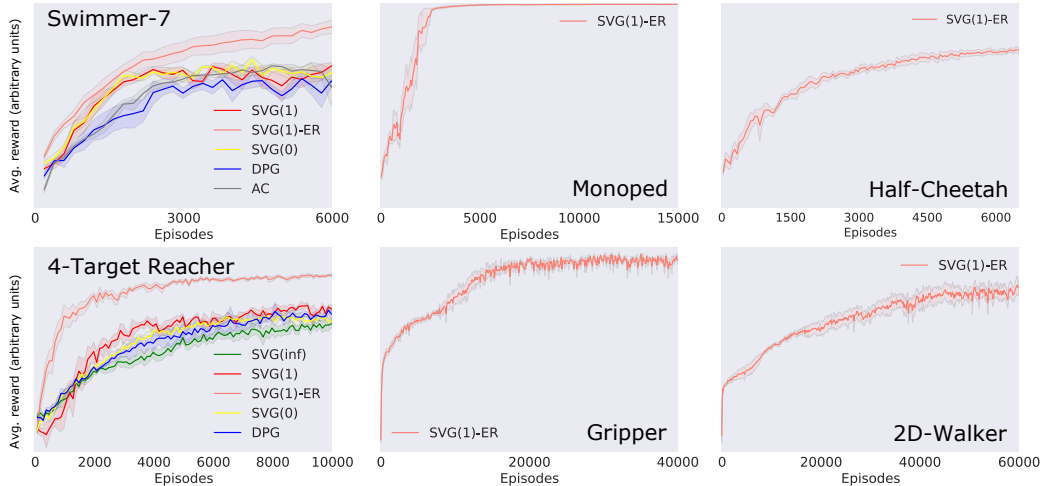

Figure 4: Across several different domains, SVG(1)-ER reliably optimizes policies, clearly settling into similar local optima. On the 4-target Reacher, SVG(1)-ER shows a noticeable efficiency and performance gain relative to the other algorithms.

higher adversarial noise levels than used during training (up to $25\%$ noise). We were able to learn gripping and walking behavior, although walking policies that achieved similar reward levels did not always exhibit equally good walking phenotypes.

# 7 Related work

Writing the noise variables as exogenous inputs to the system to allow direct differentiation with respect to the system state (equation 7) is a known device in control theory [10, 7] where the model is given analytically. The idea of using a model to optimize a parametric policy around real trajectories is presented heuristically in [17] and [1] for deterministic policies and models. Also in the limit of deterministic policies and models, the recursions we have derived in Algorithm 1 reduce to those of [2]. Werbos defines an actor-critic algorithm called Heuristic Dynamic Programming that uses a deterministic model to roll-forward one step to produce a state prediction that is evaluated by a value function [31]. Deisenroth et al. have used Gaussian process models to compute policy gradients that are sensitive to model-uncertainty [6], and Levine et al. have optimized impressive policies with the aid of a non-parametric trajectory optimizer and locally-linear models [13]. Our work in contrast has focused on using global, neural network models conjoined to value function approximators.

# 8 Discussion

We have shown that two potential problems with value gradient methods, their reliance on planning and restriction to deterministic models, can be exorcised, broadening their relevance to reinforcement learning. We have shown experimentally that the SVG framework can train neural network policies in a robust manner to solve interesting continuous control problems. The framework includes algorithm variants beyond the ones tested in this paper, for example, ones that combine a value function with $k$ steps of back-propagation through a model (SVG(k)). Augmenting SVG(1) with experience replay led to the best results, and a similar extension could be applied to any SVG(k). Furthermore, we did not harness sophisticated generative models of stochastic dynamics, but one could readily do so, presenting great room for growth.

**Acknowledgements**   We thank Arthur Guez, Danilo Rezende, Hado van Hasselt, John Schulman, Jonathan Hunt, Nando de Freitas, Martin Riedmiller, Remi Munos, Shakir Mohamed, and Theophane Weber for helpful discussions and John Schulman for sharing his walker model.

## Footnotes

[1] We make use of a time-varying reward function only in one problem to encode a terminal reward.

[2] $\gamma < 1$ for the infinite-horizon case.

[3]In the finite-horizon formulation, the gradient calculation starts at the end of the trajectory for which the only terms remaining in eq. (9) are $v^T_{\mathbf{s}} \approx r^T_{\mathbf{s}} + r^T_{\mathbf{a}}\pi^T_{\mathbf{s}}$. After the recursion, the total derivative of the value function with respect to the policy parameters is given by $v^0_{\theta}$, which is a one-sample estimate of $\nabla_{\theta}J$.

[4]Note that $\pi$ is a function of the state and noise variable.

[5]We also tested REINFORCE on this problem but achieved very poor results due to the long horizon.

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
