[Supplementary Material · paper_camera_ready2_pp-10-12.pdf]

## A  Derivation of recursive gradient of the deterministic value function

The use of derivatives in equation 4 is subtle, so we expand on the logic here. We first note that a change to the policy parameters affects the immediate action as well as each future state and action. Thus, the total derivative $\frac{d}{d\theta}$ can be expanded to

$$\frac{d}{d\theta} = \left[ \sum_{t \geq 0} \frac{d\mathbf{a}^t}{d\theta} \frac{\partial}{\partial \mathbf{a}^t} + \sum_{t > 0} \frac{d\mathbf{s}^t}{d\theta} \frac{\partial}{\partial \mathbf{s}^t} \right]$$

$$= \left( \frac{d\mathbf{a}^0}{d\theta} \frac{\partial}{\partial \mathbf{a}^0} + \frac{d\mathbf{s}^1}{d\theta} \frac{\partial}{\partial \mathbf{s}^1} \right) + \left[ \sum_{t \geq 1} \frac{d\mathbf{a}^t}{d\theta} \frac{\partial}{\partial \mathbf{a}^t} + \sum_{t > 1} \frac{d\mathbf{s}^t}{d\theta} \frac{\partial}{\partial \mathbf{s}^t} \right].$$

Let us define the operator $\nabla_\theta^t \triangleq \left[ \sum_{t' \geq t} \frac{d\mathbf{a}^{t'}}{d\theta} \frac{\partial}{\partial \mathbf{a}^{t'}} + \sum_{t' > t} \frac{d\mathbf{s}^{t'}}{d\theta} \frac{\partial}{\partial \mathbf{s}^{t'}} \right]$. The operator obeys the recursive formula

$$\nabla_\theta^t = \left( \frac{d\mathbf{a}^t}{d\theta} \frac{\partial}{\partial \mathbf{a}^t} + \frac{d\mathbf{s}^{t+1}}{d\theta} \frac{\partial}{\partial \mathbf{s}^{t+1}} \right) + \nabla_\theta^{t+1}.$$

We can transform this to

$$\nabla_\theta^t = \frac{d\mathbf{a}^t}{d\theta} \left( \frac{\partial}{\partial \mathbf{a}^t} + \frac{d\mathbf{s}^{t+1}}{d\mathbf{a}^t} \frac{\partial}{\partial \mathbf{s}^{t+1}} \right) + \nabla_\theta^{t+1}.$$

The value function depends on the policy parameters $V^t(\mathbf{s}^t; \theta)$. The deterministic Bellman equation can be specified as $V^t(\mathbf{s}^t; \theta) = r(\mathbf{s}^t, \mathbf{a}^t) + \gamma V^{t+1}(\mathbf{s}^{t+1}; \theta)$. Now, we can apply the operator $\nabla_\theta^t$:

$$\nabla_\theta^t V^t(\mathbf{s}^t; \theta) = \nabla_\theta^t \left[ r(\mathbf{s}^t, \mathbf{a}^t) + \gamma V^{t+1}(\mathbf{s}^{t+1}; \theta) \right]$$

$$= \left[ \frac{d\mathbf{a}^t}{d\theta} \left( \frac{\partial}{\partial \mathbf{a}^t} + \frac{d\mathbf{s}^{t+1}}{d\mathbf{a}^t} \frac{\partial}{\partial \mathbf{s}^{t+1}} \right) + \nabla_\theta^{t+1} \right] \left[ r(\mathbf{s}^t, \mathbf{a}^t) + \gamma V^{t+1}(\mathbf{s}^{t+1}; \theta) \right]$$

$$= \frac{d\mathbf{a}^t}{d\theta} \frac{\partial}{\partial \mathbf{a}^t} r(\mathbf{s}^t, \mathbf{a}^t) + \frac{d\mathbf{a}^t}{d\theta} \frac{d\mathbf{s}^{t+1}}{d\mathbf{a}^t} \frac{\partial}{\partial \mathbf{s}^{t+1}} \gamma V^{t+1}(\mathbf{s}^{t+1}; \theta) + \nabla_\theta^{t+1} \gamma V^{t+1}(\mathbf{s}^{t+1}; \theta).$$

In the "tick" notation of the main text, this is equation 4.

## B  Gradient calculation for noise models

Evaluating the Jacobian terms in equations in equations 9 and 10) may require knowledge of the noise variables $\eta$ and $\xi$. This poses no difficulty when we obtain trajectory samples by forward-sampling $\eta, \xi$ and computing $(\mathbf{a}, \mathbf{s}')$ using the policy and learned system model.

However, the same is not true when we sample trajectories from the real environment. Here, the noise variables are unobserved and may need to be "filled in" to evaluate the Jacobians around the right arguments.

Equations 11 and 12 arise from an application of Bayes' rule. We formally invert the forward sampling process that generates samples from the joint distribution $p(\mathbf{a}, \mathbf{s}', \eta, \xi | \mathbf{s})$. Instead of sampling $\eta, \xi \sim \rho$ and then $\mathbf{a} \sim p(\mathbf{a}|\mathbf{s}, \eta)$, $\mathbf{s}' \sim p(\cdot|\mathbf{s}, \mathbf{a}, \xi)$, we first sample $\mathbf{a} \sim p(\mathbf{a}|\mathbf{s})$ and $\mathbf{s}' \sim p(\mathbf{s}'|\mathbf{s}, \mathbf{a})$ using our policy and the real environment. Given these data and a function $\mathbf{g}$ of them, we sample $\eta, \xi \sim p(\eta, \xi | \mathbf{s}, \mathbf{a}, \mathbf{s}')$ to produce

$$\mathbb{E}_{\rho(\xi, \eta)} \mathbb{E}_{p(\mathbf{a}, \mathbf{s}'|\mathbf{s}, \xi, \eta)} \mathbf{g}(\mathbf{s}, \mathbf{a}, \mathbf{s}', \xi, \eta) = \mathbb{E}_{p(\xi, \eta, \mathbf{a}, \mathbf{s}'|\mathbf{s})} \mathbf{g}(\mathbf{s}, \mathbf{a}, \mathbf{s}', \xi, \eta)$$

$$= \mathbb{E}_{p(\mathbf{a}, \mathbf{s}'|\mathbf{s})} \mathbb{E}_{p(\xi, \eta|\mathbf{s}, \mathbf{a}, \mathbf{s}')} \mathbf{g}(\mathbf{s}, \mathbf{a}, \mathbf{s}', \xi, \eta). \tag{14}$$

For example, in eq. 11, we plug in $\mathbf{g}(\mathbf{s}, \mathbf{a}, \mathbf{s}', \xi, \eta) = r_\mathbf{s} + r_\mathbf{a} \pi_\mathbf{s} + \gamma V'_{\mathbf{s}'}(\hat{\mathbf{f}}_\mathbf{s} + \hat{\mathbf{f}}_\mathbf{a} \pi_\mathbf{s})$.

## C  Model and value learning

We found that the models exhibited the lowest per-step prediction error when the $\hat{\mu}$ and $\hat{\sigma}$ vector components were computed by parallel subnetworks, producing one $(\hat{\mu}, \hat{\sigma})$ pair for each state dimension, i.e., $[(\hat{\mu}_1, \hat{\sigma}_1); (\hat{\mu}_2, \hat{\sigma}_2); \ldots]$. (This was due to varied scaling of the dynamic range of the state dimensions.) In the experiments in this paper, the $\hat{\sigma}$ components were parametrized as constant biases per dimension. (As remarked in the main text, this implies that they do not contribute to the gradient calculation. However, in the Hand environment, the planner agent forward-samples based on the learned standard deviations.)

| **Algorithm 3** SVG(0) with Replay | **Algorithm 4** Fitted Policy Evaluation |
|---|---|
| 1: Given empty experience database $\mathcal{D}$ | 1: Given experience database $\mathcal{D}$ |
| 2: **for** $t = 0$ **to** $\infty$ **do** | 2: Given value function $\hat{V}(\cdot, \nu)$, outer loop time $t$ |
| 3:   Apply control $\pi(\mathbf{s}, \eta; \theta)$, $\eta \sim \rho(\eta)$ | 3: $\nu^{new} = \nu$ |
| 4:   Observe $r, \mathbf{s}'$ | 4: **for** $m = 0$ **to** $M$ **do** |
| 5:   Insert $(\mathbf{s}, \mathbf{a}, r, \mathbf{s}')$ into $\mathcal{D}$ | 5:   Sample $(\mathbf{s}^k, \mathbf{a}^k, r^k, \mathbf{s}^{k+1})$ from $\mathcal{D}$ ($k < t$) |
| 6:   // Critic updates | 6:   $y^m = r^k + \gamma \hat{V}(\mathbf{s}^{k+1}; \nu)$ |
| 7:   Train value function $\hat{Q}$ using $\mathcal{D}$ | 7:   $w = \frac{p(\mathbf{a}^k|\mathbf{s}^k; \theta^t)}{p(\mathbf{a}^k|\mathbf{s}^k; \theta^k)}$ |
| 8:   // Policy update | 8:   $\Delta = \nabla_{\nu^{new}} \frac{w}{2}(y^m - \hat{V}(\mathbf{s}^k; \nu^{new}))^2$ |
| 9:   Sample $(\mathbf{s}^k, \mathbf{a}^k, r^k, \mathbf{s}^{k+1}, \mathbf{a}^{k+1})$ from $\mathcal{D}$ ($k < t$) | 9:   Apply gradient-based update to $\nu^{new}$ using $\Delta$ |
| 10:   Infer $\eta^k|(\mathbf{s}^k, \mathbf{a}^k)$ | 10:   Every $C$ updates, set $\nu = \nu^{new}$ ($C = 50$) |
| 11:   $v_\theta = \hat{Q}_{\mathbf{a}} \pi_\theta\big|_{\eta^k}$ | 11: **end for** |
| 12:   Apply gradient-based update using $v_\theta$ | |
| 13: **end for** | |

# D   Experimental details

For a direct comparison we ran SVG(0), SVG(1), SVG($\infty$), and DPG without experience replay for the policy updates since replay is not possible for SVG($\infty$). (All algorithms use replay for the value function updates.) We further implemented SVG(1)-ER such that policy updates are performed at the end of each episode, following the update of the value function, rather than following each step of interaction with the environment. For $K$ steps of replay for the policy we perform $K$ gradient steps according to lines 10-14 in Algorithm 2, drawing new data from the database $\mathcal{D}$ in each step. In some cases we found it helpful to apply an additional regularizer that penalized $\mathbb{E}_{\mathbf{s}}\left[\mathcal{D}_{\mathrm{KL}}[\pi_{\theta^{k-1}} || \pi_\theta]\right]$ during each step of replay, where $\pi_\theta$ denotes the policy at the beginning of the update and $\pi_{\theta^{k-1}}$ the policy after $k - 1$ steps of replay. The expectation with respect to $\mathbf{s}$ is taken over the empirical distribution of states in $\mathcal{D}$. Following [30] we truncated importance weights (using a maximum value of 5 for all experiments except for the 4-target reacher and gripper where we used 20).

Computing the policy gradient with SVG($\infty$) involves backpropagation through what is effectively a recurrent network. (Each time-step involves a concatenation of two multi-layer networks for policy and dynamics model. The full model is a chain of these pairs.) In line with findings for training standard recurrent networks, we found that "clipping" the policy gradient improved stability. Following [19] we chose to limit the norm of the policy gradient; i.e., we performed backpropagation for an episode as described above and then renormalized $v_\theta^0$ if its norm exceeded a set threshold $V_\theta^{\max}$: $\tilde{v}_\theta^0 = \frac{v_\theta^0}{\|v_\theta^0\|} \min(V_\theta^{\max}, \|v_\theta^0\|)$. We used this approach for all algorithms although it was primarily necessary for SVG($\infty$).

We optimized hyper-parameters separately for each algorithm by first identifying a reasonable range for the relevant hyper-parameters and then performing a more systematic grid search. The main parameters optimized were: learning rates for policy, value function, and model (as applicable); the number of updates per episode for policy, value function, and model (as applicable); regularization as described above for SVG(1)-ER; $V_\theta^{\max}$; the standard deviation of the Gaussian policy noise.

**Hand**

$$r^t(\mathbf{s}, \mathbf{a}) = \begin{cases} \alpha_1 ||\mathbf{a}||^2 & t < 10s, \\ \alpha_2[d(\text{hand}, 0) + d(\text{ball}, \text{target})] & t = 10s, \end{cases} \tag{15}$$

for Euclidean distance $d(\cdot, \cdot)$.

**Swimmer**   $r(\mathbf{s}, \mathbf{a}) = \alpha_1 \hat{\mathbf{r}}_{\text{goal}} \cdot \mathbf{v}_{\text{c.o.m}} + \alpha_2 ||\mathbf{a}||^2$, where $\hat{\mathbf{r}}_{goal}$ is a unit vector pointing from the nose to the goal.

For the MuJoCo environment problems, we provide .xml task descriptions.

**Discount factors**   Hand: 1.0; Swimmer: 0.995; Reacher: 0.98; Gripper: 0.98; Hopper: 0.95; Cheetah: 0.98; Walker: 0.98

Table 1: Sizes of Network Hidden Layers.

| TASK | POLICY | VALUE FUNCTION | MODEL[*] |
|---|---|---|---|
| HAND | 100/100 | N/A | 20/20 |
| CARTPOLE | 100/100 | 200/100† | 20/20 |
| SWIMMER 3 | 50/50 | 200/100 | 20/20† |
| SWIMMER 5 | 100/100 | 200/100 | 20/20 |
| SWIMMER 7 | 100/100 | 200/100 | 40/40 |
| REACHER | 100/100 | 400/200 | 40/40 |
| MONOPED | 100/100 | 400/200 | 50/50 |
| CHEETAH | 100/100 | 400/200 | 40/40 |

[*] We use one sub-network of this many hidden units *per* state-dimension. † With the exception of the results in Figures 2 and 3, where the sizes are given.