[Reviews · NeurIPS 2015]

Submitted by Assigned_Reviewer_1

[Light review]

This is an interesting work that proposes some value gradient methods for stochastic systems.

The work does not have any theoretical guarantee, but claimed are supported by empirical studies.

Some aspects of the papers are not very clear though:

* The discussion around equations (12-13), and especially how to generate samples from p(eta,zeta|s,a,s') is not clear. What does "without loss of generality" mean at Lines 187-188? * How difficult is it to learn such a noise model? Generating samples from p(eta,zeta|s,a,s') is required at line 11 of Algorithm 1, line 12 of Algorithm 2, so a more clear discussion is needed.

* Why do we have partial derivative of V(s') w.r.t. theta in equation (4)? Isn't that derivative equal to V_{s'}(s') f_a pi_theta, which is there already?

* Please provide a better comparison with the work of Fairbanks. Is the main difference in the use of re-parameterization technique?

===== [UPDATE] Thank you. You addressed my concerns.
Summary: See comments.

Submitted by Assigned_Reviewer_2

The paper presents a new stochastic value gradient algorithm that can combine learning a system dynamics model with learning a (state-action) value function to obtain accurate gradients for the policy update. Value gradient algorithms could so far be used only for deterministic environments and deterministic policies. They extend the formulation to the stochastic case by introducing the noise as additional variables that are supposed to be known for the trajectories (and hence, deterministic). Furthermore, they propose to use a learned model (e.g. a neural network) to compute the gradient from multi-step predictions. Yet, the model is used only for computing the gradient and not for the prediction itself (real roleouts are used), which makes the approach less reliant on the model accuracy (in contrast to other model-based RL methods such as PILCO that suffer from model errors quite severly). They introduce 3 algorithms: SVG(0) uses the Q-function to compute the policy gradient, SVG(1) uses the value function and a 1step prediction with the learned model and SVG(inf) only uses the model without any value function. SVG(1) is reported to work best in the experiments.

There is not much to say about this paper. Nice ideas, properly executed with a massive amount of evaluations and well written. I think the contributions of the paper are very relevant as it nicely combines model-free and model-based RL methods in one framework and can be applied for complex neural network policies, value functions and dynamic models. The introduced extensions to standard value gradients such as stochastic policies and models, using the model only for gradient computation and not for prediction as well as experience replay with off-policy samples are well motivated and important contributions to make the algorithm more efficient.

More Comments:

-The algorithm SVG(0) seems to be almost equivalent to the submission 1107 (deep deterministic policy gradient). I assume the authors of the papers are the same. Please elobarate on the difference of the two algorithms and cite the other paper if both are accepted.

- The authors do not seem to update the exploration policy to generate the exploration noise usde for gathering data. This seems to be a bigger limitation of the approach that the exploration noise variance can not be learned, which might lead to more efficient exploration strategies.

Summary: A nice paper introducing a new class of algorithms that can learn the model, value function and policy simultanously. The results are exhaustive and convincing.

Submitted by Assigned_Reviewer_3

This paper introduces a family of Stochastic Value Gradient (SVG) algorithms for finding optimal policies in Markov Decision Processes. First, the authors present a standard value gradient method used for deterministic systems. The method consists in deriving the gradient of the value function (Bellman equation) with respect to the policy parameters, and backpropagating the gradient to the first time-step. To generalize this technique to stochastic systems, the authors focus on a particular type of transition functions (dynamics) where the next state is given by a deterministic function of the previous state and action, and a random noise. The noise is an exogenous variable that is assumed to be independent of the states and actions. By containing the stochasticity in a variable that is independent of states and actions, one can use a simple MC estimator to calculate the value gradient, which is computed in the same way as in a deterministic system. The gradient can be computed by fitting a model and a value function. The authors demonstrate their approach in a number of simulated systems.

The paper is very clear, well-organized and well-written. Some parts are unnecessarily long, such as the first paragraph of Section 4. Some statements seem to be repeated or unnecessary, such as 173-177. The statement about "the first in the literature to learn a model, a value function and a policy simultaneously in a continuous domain" is too strong. I can think of several works that did that, e.g., Deisenroth's works.

My main concern with this work is the fact that the derivation of the SVG algorithms is rather straightforward. The re-parameterization technique is a simple trick that can be applied only for certain types of transition functions (state-independent noise). The importance-sampling technique is extensively used in off-policy reinforcement learning. Of course, the simplicity of the derivations should never be used as an argument against a paper if the authors clearly demonstrate the advantage of their approach on real-world problems. The experiments presented in this paper are limited to toy problems, which does not indicate how much impact this work will have. Moreover, it is not very clear from Figures 2-4 which variant of SVG should be used.

On the other hand, I really liked the clarity of the paper, and I can easily see how other researchers may attempt to implement the SVG algorithms and reproduce the results. This is a big plus for the paper. The authors also carefully evaluated their algorithms on a quite large number of systems.

Some questions/comments: 1) In algorithms 1 and 2, there is a bias in the choice of actions caused by the policy parameters. How do you remove this bias in training the model? The model is tested on data points that are sampled from a distribution (policy) that is different from the distribution used in the training.

2) There seems to be an error in line 11 of Algorithm 2. The numerator should be (s,a) instead (s^k,a^k). 3) Could you detail the derivation of Equations (3) and (4)? 4) Typo: "that that".
Summary: This is well-written paper. The idea of Stochastic Value Gradient is nice but it seems quite straightforward, and all the experiments were performed on artificial problems.

Submitted by Assigned_Reviewer_4

The paper introduces a formalizm which enables that allows extending value-based

policy gradient methods to stochastic policies in stochastic

environments. The formalism is based on reparametrization (modeling

stochasticity as a deterministic function of exogenuous noise).

The paper presents algorithms based on this formalism and test them on

multiple control problems.

The problem of extending value-gradient methods to stochastic policies

and environments seems important. The experimental part of the paper

covers different challenging control domains. The main issue with the

paper is that the presentation needs significant improvement to be accessible

to a general RL audience, which is a reasonable requirement from an RL

paper. Specifically, terms are used without being defined first, many

descriptions are informal where formality is required, and many critical

details are missing.

This makes it hard to follow and evaluate the exact

contributions of the paper for people outside the community that works on

RL for robotics in continuous state/action spaces.

Details:

- the abstract is currently a bit misleading: it claims to apply

algorithms to "a range of difficult physical control problems", without

mentioning that they are *simulated* control problems. The authors should

clarify this point explicitely in the abstract in a final version of the

paper, if accepted.

line 16 - long sentence, needs break.

line 38 - what do you mean by 'chain' - please be more formal

line 40 - 'backpropagating' has different meanings - please

explain which one of them you refer to here. What is being backpropagated?

line 41 - 'replace the derivatives' - replace them where?

line 51, 55 - what do you mean by 'unroll', please clarify

line 56 - 'learned jointly' - why is it new, many model-based

algorithms can be considered to learn jointly the model,

value-function and policy

line 57 - long sentence, break

line 60-61 - the sentence is vague - please be more explicit

line 76 - Define MDP as a tuple (S, A, P, R, gamma, s0)

line 78 - please clarify why learning time-invariant policies if

reward is time-varying

line 82 - define gamma mathematically (instead of 'strictly less than...')

Equation 1 - missing upper limit in the summation

line 92-94 - definition of V' is confusing, also you might want to use ':="

line 105 - you differentiate with respect to state and theta (not action)

Eq. 3-4 - please provide more details on how you perform the derivation here

line 108 - 'purely model-based formalism' expression is used but was never defined here

line 108 - earlier, did you mean by 'backpropagating' to

backpropagation of the derivatives described here? if yes,

please say explicitely

line 111 - 'unrolling' was never defined

lines 111-114 are unclear

line 117 - 'model' used but never defined

line 118 - how do you define 'modeling errors' and how

sensitive V_theta to them ('highly sensitive' is not informative

enough)

line 119 - 'we start instead' - instead of what?

line 122 - 'real, not planned' - what do you mean by 'planned' - please define

line 124 - 'critics' used but never defined in this context

lines 125-127 - sentence is unclear

line 131 - 'backpropagate efficiently' - what do you mean by saying

that: what is your efficiency measure

line 136 - 'we consider conditional densities' - what do you mean by 'consider'

line 148 - what do you mean by 'exploits' the Jacobian

line 165 - 'standard planning method' - can you provide reference?

please define 'planning' in this context?

Eq 10-11 - you use notation that is not explained: what is '|_\eta,\ksi'

line 185 - 'we can now replace..' - why you can do that?

line 186 - why this case is 'important'

line 187 - \Sigma was not defined

Sec 4.2 - Since it doesn't contain any details, consider moving it to

forward near the place you provide more details

line 218 - 'the parameters of the critic' used but were never defined here

line 257 - 'our framework easily supports' - what do you mean by 'easily'

Sec 6 - it would be good to mention a relevant figure first (e.g. Fig 2) and

then describe in detail what it shows.

Section 6.1 - based on what you determine that [25] is

state-of-the-art? also is it state-of-the-art on a given domain?

multiple domains?

Summary: The paper addresses an interesting problem, presents new formalism and

algorithms, and test them extensively on multiple control problems.

In its current state, the paper seems to be accessible only to a

sub-community of RL. To make the paper more widely accessible (at least

to a general RL audience), and to increase the impact of the paper, the

presentation needs to be improved - see detailed comments below.

Also, the abstract is currently a bit misleading: it claims to apply

algorithms to "a range of difficult physical control problems", without

mentioning that they are *simulated* control problems. The authors should

clarify this point explicitely in the abstract in a final version of the

paper, if accepted.

Author Feedback
Author rebuttal: We thank all reviewers for taking time to understand the manuscript and provide detailed feedback, which will greatly improve the manuscript. We are happy to see agreement that the paper addresses an important set of problems, introduces useful ideas and algorithms, and presents a thorough experimental investigation on several domains.

Below we address specific points:

[R2] Significance of re-parameterization; importance sampling novelty (IS); task complexity; joint learning
Re-parameterization may be simple but it is a powerful idea that has only recently found wider application in ML. Importantly, the application here leads to a new perspective on policy gradients (i.a. avoiding likelihood ratio estimators; allowing backprop through stoch. model); for example, the recently published DPG algorithm follows as a special case of reparameterization. We don't claim IS is novel but use it to construct new, practical algorithms.

Our experiments are performed in simulation, but with complexity on par with recent work (e.g. ref [11]), and include locomotion, reaching, and grasping in high dimensions with realistic, nonlinear contact physics.

PILCO does not use value function approximators.

[R2,R3,R5] Derivation of eq. 3,4, extra term: (3,4) result from differentiating the recursive definition of the value function (line 103) wrt. s and \theta, taking into account that V(s') depends on \theta through s' (line 104; 2nd term in (4)) but also directly, since the policy parameters affect future time steps (3rd term in (4)).

[R4,R5] Inferring noise variables in (12,13); stochastic neural networks (NN)
We choose the parametric form for policy and model and thus know the form of the joint p(s',a, \xi, \eta | s ). We can then perform inference via Bayes rule. Depending on the parametrization this is easy or requires approximation. For the additive-Gaussian noise considered in the paper \xi, \eta are deterministic functions of (s, s', a).

To parameterize conditional densities for model and policy we use the form in l134 (general case), and l259/265 (for the model). We use NNs to learn \mu and learn a constant sigma per dimension. \rho is std-normal. The NNs hence parameterize the mean \mu of a Gaussian as a function of state (and action). Supervised learning for such a model is straightforward. More general noise models are possible (see below).

[R2] State dependent noise
Fixed noise was sufficient for our experiments but reparameterization admits state-dependent noise as discussed in paper lines 256ff (e.g. learning sigmas as NNs). More general transformations allow more complex forms of noise (see below)

[R1] Learning exploration
SVG may learn policy noise although we are not using this here. The policy noise can be state dependent (as in line 134). We can learn richer stochastic policies by feeding noise into the network. For instance, we have used SVG(0) to learn a bimodal action distribution for a partially observed domain in which bimodality is optimal.

[R2] Model bias
Training the model concurrently with the policy ensures that it is updated as the agent enters new parts of state space. Exploration ensures that actions are well sampled around trajectories (maintaining exploration is critical). We compute policy gradients on observed trajectories to ensure they are evaluated only where model training data is available.

[R2] Alg. 2,L11: We importance weight the historical state-action pair (s_k, a_k) from the database.

[R5] "without loss of generality" in lines 187-188: State-independent additive noise does not affect the model Jacobian.

[R1] SVG(0)/DPG
DPG is the deterministic policy limit of SVG(0). Hence their Q functions are different. Both allow off-policy learning and experience replay but in different ways. Only SVG(0) allows the off-policy use of K-step returns for learning Q, or learning exploration.

[R2] Which SVG
We found that SVG(1)-ER performed very well across all problems except for the "Hand" due to its long horizon and sparse reward (Fig. 2, left). SVG-variants with value fn failed, but purely model-based SVG(inf) succeeded.

[R5] Comparison w/ Fairbank: Fairbank assumes known, deterministic dynamics. We extend to the stochastic case, show how to robustly combine approx. models and value fns., and tackle challenging problems.

[R4] comparison w/other algorithms (PILCO; Levine)
PILCO is limited to linear/RBF policies. To the extent that the environments are comparable, our results for the length 3 swimmer are similar to Levine & Abbeel 2014.

[R4] Reproducibility: We will include a full description of all domains in the final version of the paper.

[R3] Line 185: We model the joint trajectory distribution and condition on observed data.
[R3] Line 40/108: Yes, see eqns. 3&4.
[R3] Line 148: The likelihood ratio estimator doesn't use the Jacobian of g. E.g. D2 in ref [16]
[R3] Line 122: Planned = trajectory sampled from learned model.